# Refining Bias and Reward in LLM Recommender Agents through Meta-Controlled Tool Invocation

## Abstract

Large language model (LLM) agents have recently been brought to recommender systems given their flexible capability of tool use. Although existing approaches adopt the reasoning and acting paradigms for profiling, planning, and memory augmentation, they remain ad hoc and overlook core recommendation challenges in agent-environment interactions, including debiasing and reward estimation in offline learning scenarios. In this paper, we introduce BARO (Bias And Reward Optimization), a meta-controlled, tool-augmented LLM agent framework that explicitly addresses these challenges. BARO employs a two-stage recommendation process: a coarse recommender generates a candidate slate based on user history, and a meta-controller adaptively invokes three specialized tools to refine the recommendation results: a bias detector assesses and mitigates bias in the candidate set, a reward estimator calibrates noisy offline rewards, and an action grounder selects final recommendations from the candidate pool. This design injects bias correction and reward refinement directly into the agent's decision loop in the recommendations. Empirical results on two benchmark datasets demonstrate that BARO achieves consistent improvements over state-of-the-art methods in metrics such as accuracy, diversity, and fairness. The code will be made publicly available upon acceptance.

## 1 Introduction

Recommender systems (RecSys) are the cornerstone of modern digital platforms, such as e-commerce, streaming services, and short video applications (Bobadilla et al., 2013; Burke, 2002), where a key task is to align recommendations with user preferences and improve long-term user satisfaction (Wang et al., 2024a). In modern RecSys, semantic information about users and items has been a key factor in effective recommendations, for example, the profile of a user or the genre of a movie (Harper & Konstan, 2015). However, how to meaningfully leverage this information remains a challenge.

Recently, large language models (LLMs), with their remarkable contextual understanding and reasoning abilities (Touvron et al., 2023; Guo et al., 2025), have been applied to RecSys to capture complex patterns between users, items, and their interactions. These methods can be grouped into three paradigms (Wu et al., 2024). (1) In-context learning (ICL) approaches (Dong et al., 2024) treat LLMs as powerful general purpose functions, extracting user preferences via prompt engineering without modifying model parameters (Liu et al., 2023). Although convenient, ICL methods inherit hallucinations and biases from general-purpose LLMs, since these models are not built for recommendation-specific tasks (Gao et al., 2025a; Cai et al., 2025a). (2) Post-training approaches (Gao et al., 2025b; Hu et al., 2022) fine-tune LLMs in RecSys tasks to alleviate hallucination and bias issues. However, post-training is computationally costly, and the resulting models often lack extensibility to new domains or objectives (Mohammadi et al., 2025; Cai et al., 2025b; Huang et al., 2025). (3) In contrast, LLM agents (Wang et al., 2025b; Ning et al., 2024; Wang et al., 2024b) further extend flexibility by enabling tool use, allowing recommendation pipelines to integrate auxiliary

modules beyond static LLM predictions. This flexibility of LLM agents has motivated a surge of effective agent frameworks for RecSys. One branch of work (Zhang et al., 2024c; Shi et al., 2024b) optimizes a memory module to capture long-term engagement, while others (Zhang et al., 2024a; Gong et al., 2024) strengthen planning capabilities to enhance reasoning before micro-level recommendations.

Despite the promise of LLM agents in RecSys, most of these agentic frameworks were not designed specifically for RecSys, leading to overlooking the interaction bias and the user interaction offline reward issues. Generally, a recommendation process is framed as a sequential decision-making problem, where user preference can be viewed as implicit rewards (Chen et al., 2023b;a; Xue et al., 2023). Yet current approaches focus mostly on agent memory and planning via prompting the LLMs, in which **biases** and **reward inaccuracies** in offline environments remain largely unaddressed. These issues are critical: biases such as popularity bias or filter bubbles (Gao et al., 2023a;b) can harm fairness and diversity, while inaccurate reward estimation can mislead optimization. Moreover, integrating LLM agents into RecSys introduces additional challenges, including sycophantic behaviors (Pan et al., 2024) and hallucination risks, which exacerbate the inherent bias and reliability concerns.

To tackle these challenges, we propose **BARO** (*Bias And Reward Optimization*), a meta-controlled, tool-augmented LLM agent specifically designed for recommendations. BARO adopts a two-stage recommendation pipeline. First, an in-context learning-based coarse recommender produces candidate slates from user interaction histories. Then, a **Meta-Controller** governs tool use to refine the slate, invoking three specialized modules: (i) a **Bias Detector**, which assesses the bias level of the candidate set and triggers regeneration when biases exceed thresholds; (ii) a **Reward Estimator**, which evaluates uncertainty and corrects noisy offline rewards; and (iii) an **Action Grounder**, which integrates information from all modules to make fine-grained final recommendations. By explicitly embedding debiasing and reward refinement into the decision-making loop, BARO provides a RecSys-specific design that moves beyond generic reasoning and acting agents, yielding state-of-the-art performance on reliable, fair, and accurate recommendations with evaluations on two widely used benchmarks. Our contributions are threefold:

- We highlight the limitations of existing ReAct-style LLM agents in recommendation and formulate the overlooked challenges of biases and reward inaccuracies in offline environments, motivating a RecSys-specific agent framework.

- We propose BARO (*Bias And Reward Optimization*), a meta-controlled, tool-augmented LLM agent. BARO integrates a Bias Detector, Reward Estimator, and Action Grounder under a meta-controller, enabling explicit debiasing and reward calibration within a two-stage pipeline.

- We conduct experiments on two benchmark datasets, comparing BARO against both traditional sequential recommenders and SOTA LLM-based methods. Results show consistent improvements across diverse metrics, demonstrating the effectiveness and generality of our approach.

## 2 RELATED WORK

**Large Language Models for Recommendations.** Recent LLMs have been integrated into recommendation systems and show remarkable potential in downstream tasks (Chen et al., 2025; Bao et al., 2023a; Li et al., 2024). Early studies explore in-context learning to leverage LLMs for recommendation tasks without parameter updates (Bao et al., 2025; Liu et al., 2023; Bao et al., 2023a; Shi et al., 2024b; Gao et al., 2023c; Ye et al., 2024; Shi et al., 2024a; Bao et al., 2023b; Liao et al., 2023; Lin et al., 2024). Yet ICL-based methods suffer from hallucination and bias from their pretraining knowledge (Gao et al., 2025a; Cai et al., 2025a), and fine-tuning methods lack flexibility in adapting to new tasks (Mohammadi et al., 2025). BARO alleviates these issues by integrating LLM agents with a meta-controller, enabling more accurate recommendations under biased and implicit feedback.

**Large Language Model Agents.** Recent LLM-based agents with autonomous decision-making processes have been used to tackle complex tasks (Wang et al., 2025b; Ning et al., 2024; Cai et al., 2025a). LLM agents

can dynamically explore complex user preferences and perform multi-step planning to generate personalized recommendations (Li et al., 2024; Bao et al., 2023a; Ning et al., 2024; Huang et al., 2025; Zhao et al., 2024; Wang et al., 2025b; 2024b; Zhang et al., 2024b; Shu et al., 2024; Zhang et al., 2024a;c). Yet these existing methods often overlook key challenges such as bias and reward inaccuracies. Our proposed BARO introduces a RecSys-specific agent design with dedicated Bias Detector and Reward Estimator modules, providing accuracy, fairness, and robustness in recommendations.

## 3 PRELIMINARIES

**Model-based Offline Reinforcement Learning.** Reinforcement learning aims to learn policies for solving sequential decision-making problems in Markov Decision Processes (MDPs) (Sutton & Barto, 2018). An MDP is formulated as a 5-tuple $< \mathbf{S}, \mathbf{A}, T, R, \gamma >$, where $\mathbf{S}$ denotes the state space and $\mathbf{A}$ represents the set of feasible actions. At timestep $t$, after taking an action $\mathbf{a_t}$ in state $\mathbf{s_t}$, the system transitions to the next state $\mathbf{s_{t+1}}$ and receives a reward $r_t$. These dynamics are captured by the transition function $T$, while the reward is determined by the reward function $R$. The objective of RL is to maximize the discounted cumulative reward $G = \sum_{t=0}^{\infty} \gamma^t r_t$, where $\gamma \leq 1$ is the discount factor that balances immediate and future rewards.

While RL typically learns policies through extensive interactions with environments, offline RL instead derives policies solely from pre-collected logs, mitigating the issue of costly interactions in applications such as healthcare and recommender systems (Yu et al., 2024; Chen et al., 2023a). Denoting the offline logs as a set of transitions $\mathcal{D} = \{(s_t, a_t, s_{t+1}, r_t)\}$, model-based offline RL leverages $\mathcal{D}$ to learn a world model that serves as a proxy environment, allowing policies to be trained through interactions with this learned model. A world model usually consists of an estimated transition function, *i.e.*, $\hat{T}$, and an estimated reward function, *i.e.*, $\hat{R}$. Although model-based offline RL has demonstrated high sample efficiency and strong performance, a key challenge remains in the inaccuracy of the learned world models.

**Problem Formulation.** Due to the interactive nature and sequential decision-making paradigm, recommendation tasks can be naturally formulated as MDPs and addressed with RL (Sutton & Barto, 2018; Yu et al., 2024). A state $\mathbf{s} \in \mathbf{S}$ represents the state of the system prior to recommendation, typically including user information, (*e.g.*, recent interactions and side features) as well as item attributes. An action $\mathbf{a} \in \mathbf{A}$ corresponds to a recommended item, and the reward $r$ is derived from user feedback—for example, a movie rating on streaming platforms or the view-time ratio of a short video (Gao et al., 2022a;b). The transition function is often parameterized by sequential models such as Transformers (Vaswani et al., 2017), which encode a fixed-length history of past interactions together with the current interaction to produce the next state. Further, long-term user engagement is commonly captured by the discounted cumulative reward.

## 4 METHOD

We propose **BARO** (Bias And Reward Optimization), a meta-controlled and tool-augmented LLM agent for recommendation. BARO comprises six components: (i) a world model for simulation, (ii) an ICL-based coarse recommender, (iii) a meta-controller that orchestrates tool use, (iv) a bias detector, (v) a reward estimator, and (vi) a fine-tuned action grounder. We next describe how these modules cooperate and learn.

### 4.1 WORLD MODEL SIMULATION

We adopt a model-based offline RL setup, where a learned reward function $\hat{R}$ serves as the world model to evaluate candidate items without interacting with real users. Following prior works (Gao et al., 2023b;a; Zhang et al., 2024d), $\hat{R}$ is trained using offline logs and a supervised recommendation model, such as DeepFM (Guo et al., 2017).The reward function $\hat{R}$ is then frozen and used to provide immediate feedback for new recommendations. We do not model the transition function $\hat{T}$ explicitly, as the current interaction and history are already fed into LLM agents for reasoning.

Figure 1: The overall architecture of our proposed **BARO** framework. The process begins with a frozen Coarse Recommender generating an initial *Raw List of Items*. The core of our agent, the Meta-Controller, evaluates this list and decides whether to (1) invoke specialized tools—the Bias Detector or the Reward Estimator—for in-depth analysis, or (2) directly pass the list to the Action Grounder. If significant issues are detected by the tools, a regeneration signal (*ReGen*) prompts the Coarse Recommender to refine the list. Finally, the Action Grounder selects the *Final Item* from the satisfactory candidate list. Components marked with ♦ are trainable agents optimized via reinforcement learning, while those with ❄ are frozen.

After training the world model, following ROLeR (Zhang et al., 2024d), we can obtain a refined reward function $\widetilde{R}$. We estimate uncertainty using the absolute residual between the two rewards:

$$u(s,a) = |\widetilde{R}(s,a) - \hat{R}(s,a)|, \tag{1}$$

which captures the disagreement between the original world model and the shaped reward, serving as a proxy for reward uncertainty.

### 4.2 COARSE RECOMMENDER

The coarse recommender provides a candidate item set as a preliminary recommendation. To keep it simple, we choose an in-context learning design that employs a frozen LLM. The inputs of the coarse recommender consist of (i) recent interaction histories $h_t = [i_{t-w}, \ldots, i_{t-1}]$, where $w$ is the window size and $t$ represents the current time step; (ii) the number of regenerations (#ReGen); and (iii) the guidance, $g$, from the bias detector or the reward estimator. The output of the coarse recommender is a candidate set $\mathcal{I}_t$:

$$\mathcal{I}_t = f_{CR}(h_t, \#\text{ReGen}, g). \tag{2}$$

### 4.3 META-CONTROLLER

The meta-controller serves as the core orchestrator in BARO. It comprehensively evaluates the quality of the candidate set—including overall bias, reward estimation, and user satisfaction—and then calls the appropriate tool to explicitly mitigate the identified issue. Formally, the meta-controller takes the candidate set derived from the coarse recommender, interaction history, the number of regenerations, and some statistical information, denoted as $\mathcal{M}$, as its state and makes decisions based on that state:

$$a_{\text{meta}} = \arg\max_{a \in \mathcal{A}_{\text{meta}}} P(a|\mathcal{I}_t, h_t, \#\text{ReGen}, \mathcal{M}), \tag{3}$$

where $\mathcal{A}_{\text{meta}} = \{\text{call BD}, \text{call RE}, \text{call AG}\}$, referring to calling the bias detector, reward estimator, and action grounder, respectively. Then, we elaborate on the details of $\mathcal{M}$. Aligning with our motivation of debiasing and reward refinement, popularity bias, exposure bias, uncertainty of the reward estimation, and immediate user satisfaction are measured. To be specific, Items are categorized as popular if they fall within the top 20% of the rating distribution in the offline logs. Meanwhile, for a user's interaction history, the tags of the interacted items, *e.g.*, the movie genres, are ranked, and the top 20% are labeled as popular tags. The uncertainty estimation is calculated by Eq. 1. The immediate user satisfaction is measured by the rewards in the candidate set. To sum up, $\mathcal{M}$ is composed of the ratio of popular items $\mathcal{P}_{\text{item}}$, the ratio of items with popular tags $\mathcal{P}_{\text{tag}}$, the mean uncertainty $\bar{u}_{\mathcal{I}_t}$, and the mean reward $\bar{r}_{\mathcal{I}_t}$ of the candidate set.

If the meta-controller suspects that the candidate set is biased, it invokes the bias detector for further examination; a similar procedure is followed when reward estimation is required. Conversely, if the candidate set is deemed satisfactory, the meta-controller calls the action grounder to generate the final recommendation. In addition, the meta-controller will summarize the quality of $\mathcal{I}_t$ as $\mathcal{S}_{\text{meta}}$ for downstream modules.

$$\mathcal{S}_{\text{meta}} = f_{\text{meta}}(\mathcal{I}_t, h_t, \#\text{ReGen}, \mathcal{M}) \tag{4}$$

The meta-controller is implemented by a tunable LLM. Its training will be detailed later.

### 4.4 BIAS DETECTOR

The bias detector acts as a specialized expert for assessing the overall bias in the candidate set, *i.e.*, $\mathcal{I}_t$. Compared to the meta-controller, the bias detector has a more in-depth evaluation of the bias status. When a candidate set is suspected of bias, the bias detector is invoked. Only if severe bias is detected does the bias detector trigger the coarse recommender to regenerate a new candidate set. Its inputs include the candidate set, the summary text from the meta-controller, history interactions, interaction history, the number of regenerations, and the statistical information.

$$a_{\text{bd}} = \arg\max_{a \in \mathcal{A}} P(a|\mathcal{S}_{\text{meta}}, \mathcal{I}_t, h_t, \#\text{ReGen}, \mathcal{M}), \tag{5}$$

where $\mathcal{A} = \{\text{call AG}, \text{ReGen}\}$ and ReGen stands for regeneration. If the bias detector determines that the overall bias has a negligible impact on fine-grained recommendation, it proceeds by invoking the action grounder. Otherwise, it triggers the coarse recommender to regenerate a new candidate set. In both cases, a diagnostic summary $\mathcal{S}_{\text{bd}}$ is produced to guide the subsequent action.

$$\mathcal{S}_{\text{bd}} = f_{\text{meta}}(\mathcal{S}_{\text{meta}}, \mathcal{I}_t, h_t, \#\text{ReGen}, \mathcal{M}). \tag{6}$$

The bias detector is implemented with a frozen LLM. Although a tunable LLM could potentially enhance its performance, we adopt the frozen version for efficiency considerations.

### 4.5 REWARD ESTIMATOR

The reward estimator operates in parallel with the bias detector but serves a complementary role. Specifically, it diagnoses the overall reward status of the candidate set, focusing on both the uncertainty of reward estimates and their scale. Formally, its inputs are the same as those of the bias detector, but the summary text from the meta-controller is different.

$$a_{\text{re}} = \arg\max_{a \in \mathcal{A}} P(a|\mathcal{S}_{\text{meta}}, \mathcal{I}_t, h_t, \#\text{ReGen}, \mathcal{M}). \tag{7}$$

If the reward estimator concludes that the uncertainty and scale of the reward signals are within an acceptable range, it proceeds by invoking the action grounder. Otherwise, it calls the coarse recommender to regenerate a new candidate set. Similar to the bias detector, the reward estimator also produces a diagnostic summary $\mathcal{S}_{\text{re}}$ to guide subsequent decisions.

$$\mathcal{S}_{\text{re}} = f_{\text{meta}}(\mathcal{S}_{\text{meta}}, \mathcal{I}_t, h_t, \#\text{ReGen}, \mathcal{M}). \tag{8}$$

Similarly, the reward estimator is implemented by a frozen LLM.

### 4.6 ACTION GROUNDER

The action grounder is a fine-grained recommender responsible for selecting the final recommendation from a limited candidate set. Constraining decisions within candidate sets simplifies the recommendation process and reduces the risk of hallucination. As the final step of the recommendation, the action grounder can comprehensively consider all information from the former modules:

$$e_i = f_{\text{ag}}(\mathcal{S}, \mathcal{I}_t, h_t, \#\text{ReGen}), \tag{9}$$

where $\mathcal{S} \in \{\mathcal{S}_{\text{meta}}, \mathcal{S}_{\text{bd}}, \mathcal{S}_{\text{re}}\}$. The action grounder is implemented by a tunable LLM, and its learning is described in the next part.

### 4.7 LEARNING PIPELINE

**Training.** The training of BARO follows a two-phase pipeline. Phase I performs supervised fine-tuning warm-up for the action grounder with reward distillation from the world model under a frozen meta-controller. Phase II applies reinforcement learning on a simulated environment built from the world model.

**Phase I: Supervised Fine-Tuning (SFT).** Given a state $s_t$ (history $h_t$ and summaries $\mathcal{S}$) and the retrieved candidate set $\mathcal{I}_t$, we query the world model to obtain shaped rewards $\{\widetilde{R}(s_t, i)\}_{i \in \mathcal{I}_t}$ and uncertainties $\{u(s_t, i)\}_{i \in \mathcal{I}_t}$ (Eq. 1). We form a soft label distribution over $\mathcal{I}_t$ by temperature-scaled normalization:

$$y_i = \frac{\exp\big(\widetilde{R}(s_t, i)/\tau\big)}{\sum_{j \in \mathcal{I}_t} \exp\big(\widetilde{R}(s_t, j)/\tau\big)}, \quad i \in \mathcal{I}_t, \tag{10}$$

where $\tau > 0$ controls the sharpness. Denoting the policy of the action grounder as $\pi_\theta(\cdot \mid s_t, \mathcal{I}_t)$, we minimize a KL-style loss with uncertainty weighting:

$$\mathcal{L}_{\text{KD}}(\theta) = \mathbb{E}_{(s_t, \mathcal{I}_t)}\left[w_t \cdot \sum_{i \in \mathcal{I}_t} y_i \big(-\log \pi_\theta(i \mid s_t, \mathcal{I}_t)\big)\right], \tag{11}$$

where $w_t = \text{stopgrad}\big(\exp(-\bar{u}_{\mathcal{I}_t}/\tau_u)\big)$ downweights high-uncertainty batches. $\bar{u}_{\mathcal{I}_t}$ is the mean uncertainty on $\mathcal{I}_t$, and $\tau_u > 0$ is a temperature. This label-free SFT distills the world model into the action grounder on the retrieved candidate sets. During Phase I, both the bias detector and reward estimator use fixed thresholds. As for the meta-controller, we utilize a rule-based policy according to the statistical information $\mathcal{M}$ and keep it static. This yields a stable initialization and avoids early-stage credit assignment issues.

**Phase II: Reinforcement Learning (RL).** After SFT, we fine-tune the agent with RL on episodes that consist of up to $K$ tool invocations followed by a final selection. $K$ is set to limit the number of ReGen. At each decision step, the action grounder samples $a_t \in \mathcal{I}_t$ from $\pi_\theta$, and we obtain a shaped reward from the world model:

$$r_t^{\text{ag}} = \widetilde{r}(s_t, a_t) - \lambda_{\text{item}} \cdot \mathbb{I}\{a_t \text{ is popular}\} - \lambda_{\text{tag}} \cdot \mathbb{I}\{a_t\text{'s tag is popular}\} - \lambda_u u(s_t, a_t), \tag{12}$$

where the three coefficients control the degree of the corresponding penalties. We optimize $\pi_\theta$ to maximize the expected discounted return $\mathbb{E}[\sum_t \gamma^t r_t^{\text{ag}}]$ with PPO (Schulman et al., 2017):

$$\max_\theta \mathbb{E}\left[\min\big(\rho_t(\theta)\hat{A}_t, \text{clip}\big(\rho_t(\theta), 1 - \epsilon, 1 + \epsilon\big)\hat{A}_t\big) + \beta_{\text{ent}}\mathcal{H}(\pi_\theta)\right], \tag{13}$$

where $\rho_t(\theta) = \frac{\pi_\theta(a_t|s_t, \mathcal{I}_t)}{\pi_{\theta_{\text{old}}}(a_t|s_t, \mathcal{I}_t)}$. $\hat{A}_t$ is the GAE advantage under $r_t^{\text{ag}}$. $\epsilon$ is the clipping parameter. $\mathcal{H}(\pi_\theta)$ and $\beta_{\text{ent}}$ are the entropy penalty and its coefficient, respectively.

Training the meta-controller with sparse rewards may be insufficient, as the credit assignment problem hampers effective cooperation between the meta-controller and the action grounder. We therefore adopt a dense reward design that scores improvements in candidate quality when the meta-controller calls a tool, either BD, RE, or AG. Let $\mathcal{M}_t = (\mathcal{P}_{\text{item}}^t, \mathcal{P}_{\text{tag}}^t, \bar{u}_{\mathcal{I}_t}, \bar{r}_{\mathcal{I}_t})$ be the diagnostics computed on $\mathcal{I}_t$ (ratios of popular items/tags, mean uncertainty from Eq. 1, and mean reward). After the meta action transforms $\mathcal{I}_t \to \mathcal{I}_{t+1}$, we define

$$r_t^{\text{meta}} = w_r\big(\bar{r}_{\mathcal{I}_{t+1}} - \bar{r}_{\mathcal{I}_t}\big) - w_p\big[(\mathcal{P}_{\text{item}}^{t+1} - \mathcal{P}_{\text{item}}^t) + (\mathcal{P}_{\text{tag}}^{t+1} - \mathcal{P}_{\text{tag}}^t)\big] - w_u\big(\bar{u}_{\mathcal{I}_{t+1}} - \bar{u}_{\mathcal{I}_t}\big) - c_{\text{regen}} \cdot \mathbb{I}\{\text{ReGen}\}, \tag{14}$$

where $w_r, w_p, w_u \geq 0$ weight improvements in mean reward, reductions in popularity bias, and reductions in uncertainty, and $c_{\text{regen}} > 0$ penalizes unnecessary regenerations. The meta-controller policy $\pi_\psi^{\text{meta}}(a \mid \mathcal{I}_t, h_t, \#\text{ReGen}, \mathcal{M}_t)$ (parameterized by $\psi$) is also optimized by PPO, similar to the optimization in Eq. 12.

In practice, we alternate updates: fix $\pi_\psi^{\text{meta}}$ and optimize $\pi_\theta$ for a few epochs, then fix $\pi_\theta$ and optimize $\pi_\psi^{\text{meta}}$, and repeat this process until the predefined number of epochs is reached. We adopt parameter-efficient fine-

tuning via LoRA (Hu et al., 2022). During both SFT and RL, only the low-rank adapters are updated, while the base LLM remains frozen. Since BARO is evaluated on proactive recommendation tasks (Zhu et al., 2023), the reshaped rewards in Eq. 12 and Eq. 14 are further refined by incorporating the consecutive reward difference and the $L_2$ distance to the target items as in T-PRA (Wang et al., 2025b).

**Inference.** At inference time, BARO performs a single forward pass with a budget of at most $K$ tool calls. Given $h_t$, the coarse recommender produces $\mathcal{I}_t$. The meta-controller inspects $\mathcal{M}_t$ and either (i) invokes `BD`/`RE` to refine $\mathcal{I}_t$ and updates the summary, or (ii) directly calls the action grounder. The action grounder then selects the final item from the (refined) $\mathcal{I}_t$. All summaries ($\mathcal{S}_{\text{meta}}, \mathcal{S}_{\text{bd}}, \mathcal{S}_{\text{re}}$) are appended to prompts to ensure consistency and interpretability.

## 5 EXPERIMENTS

**Datasets and Tasks.** Two datasets, specifically Steam (Kang & McAuley, 2018) and Amazon-book (Ni et al., 2019), are used in our experiments. The statistical details of these datasets are presented in Table 1. The reward scale for both datasets is structured range from 1 to 5. The Steam dataset, sourced from a gaming platform, measures reward signals based on playtime, whereas the Amazon-Book dataset originates from a book rating platform, with user ratings function-

Table 1: Statistical overview of the Steam and Amazon-book datasets.

| Dataset | Usage | # Users | # Items | Density |
|---|---|---|---|---|
| Steam | Train | 6012 | 190365 | 0.145% |
| | Test | 6012 | 190365 | 0.084% |
| Amazon | Train | 3109 | 13864 | 0.788% |
| | Test | 3109 | 13864 | 0.320% |

ing as reward signals. In accordance with the methodology established by T-PRA (Wang et al., 2025b) to ensure a fair comparison, each dataset is partitioned into 80% for training and 20% for testing.

The proactive recommendation task, proposed by (Zhu et al., 2023), assigns each user a target item. The policy must gradually guide the user toward ultimately selecting this target item.

**Baselines.** The baseline methods fall into two categories: (i) one naive baseline and four sequential recommendation baselines, including **POP**: straightforwardly recommends the most popular items at each step; **Caser** (Tang & Wang, 2018): utilizes convolutional filters to extract sequential patterns as local features; **GRU4Rec** (Jannach & Ludewig, 2017) and **SASRec** (Kang & McAuley, 2018): exploit GRU and Transformer to encode sequential features, respectively. (ii) LLM-based recommendation baselines: **IRS** (Zhu et al., 2023): combines Transformer with impressionability mask to implement personalized proactive recommendation; **BiLLP** (Shi et al., 2024b): adopts a two-level decision-making paradigm, forming a ReACT-like framework; **LLM-IPP/LLM-IPP (CoT)/LLM-IPP (ToT)** (Wang et al., 2025a): makes use of in-context learning to handle proactive recommendation. The *CoT* and *ToT* variants are empowered by Chain-of-Thought (Wei et al., 2022) and Tree-of-Thought (Yao et al., 2023), respectively; T-PRA (Wang et al., 2025b): an LLM agent designed for proactive recommendation. It adopts DPO (Rafailov et al., 2023) to tune the adapters for enhanced performance and achieves state-of-the-art (SOTA).

**Metrics and Evaluation.** We follow the standard evaluation protocol in proactive recommendation following (Zhu et al., 2023; Wang et al., 2025a) and report two groups of metrics.

**Simulator-based metrics.** An independently trained recommender is used as a user simulator. Based on this simulator, we compute: (i) *Increment of Interest (IoI)* — the increase of the predicted score for the target item after the whole trajectory; (ii) *Increment of Rank (IoR)* — the improvement of the target item's ranking position; (iii) *Accuracy (Acc)* — the proportion of recommended items that fall within the top-$K$ positions of the simulator ranking where $K$ is a threshold and set as 50.

**LLM-based metrics.** A frozen LLM evaluator is employed to assess the recommendation sequence. Two aspects are measured: (i) *Acceptance*, the average probability that the user would accept each recommended item; (ii) *Coherence*, the average consistency between successive items in the trajectory.

All compared methods are evaluated with the same simulator and evaluator settings to ensure fairness.

Table 2: Performance comparison on Steam and Amazon Book datasets. Best results are in bold.

| Method | Steam | | | | | Amazon Book | | | | |
|---|---|---|---|---|---|---|---|---|---|---|
| | IoI | IoR | Acc. | Acce. | Coh. | IoI | IoR | Acc. | Acce. | Coh. |
| POP | -0.426 | 41.8 | 0.097 | 0.381 | 0.482 | 0.494 | -205.7 | 0.085 | 0.428 | 0.595 |
| Caser | 0.269 | 89.9 | **0.977** | 0.505 | 0.239 | 0.263 | 317.1 | **0.969** | 0.446 | 0.629 |
| GRU4Rec | 0.220 | 60.8 | 0.969 | 0.511 | 0.240 | 0.672 | 725.6 | 0.956 | 0.443 | **0.645** |
| SASRec | 0.354 | -75.14 | 0.484 | 0.259 | 0.257 | 1.060 | 436.8 | 0.563 | 0.438 | 0.506 |
| IRS | 0.164 | 218.7 | 0.934 | 0.381 | 0.249 | 0.097 | 166.4 | 0.883 | 0.470 | 0.481 |
| BiLLP | 0.427 | 308.7 | 0.855 | 0.523 | 0.477 | 1.200 | 655.3 | 0.490 | 0.583 | 0.604 |
| LLM-IPP | 0.259 | 340.6 | 0.912 | 0.651 | 0.597 | 1.436 | 845.6 | 0.844 | 0.595 | 0.557 |
| LLM-IPP (CoT) | 0.264 | 303.0 | 0.906 | 0.629 | 0.580 | 1.277 | 803.5 | 0.836 | 0.601 | 0.538 |
| LLM-IPP (ToT) | 0.282 | 244.1 | 0.861 | 0.571 | 0.509 | 0.944 | 513.0 | 0.765 | 0.522 | 0.533 |
| T-PRA | 0.584 | 432.8 | 0.894 | 0.588 | 0.403 | 1.783 | 1276.5 | 0.773 | 0.589 | 0.629 |
| BARO (Ours) | **0.601** | **469.4** | 0.886 | **0.664** | **0.601** | **1.976** | **1324.2** | 0.802 | **0.611** | 0.617 |

**Implementation Details.** In BARO, we employ Llama-3.1-8B-Instruct (Dubey et al., 2024) as the backbone for all LLM modules to ensure a fair comparison with T-PRA. Supervised fine-tuning is performed for 3 epochs, followed by reinforcement learning for another 5 epochs, where each epoch alternates between updating the meta-controller and the action grounder. We adopt the AdamW optimizer with a learning rate of $5 \times 10^{-5}$. A single run of BARO can be reproduced on two NVIDIA RTX$^{\text{TM}}$ A6000 GPUs (48 GB GDDR6) and requires approximately 30 GPU hours.

## 5.1 OVERALL

According to Table 2, our proposed method **BARO significantly outperforms both traditional and LLM-based baselines** in terms of the core metrics IoI and IoR compared to the strongest baseline T-PRA. Specifically, BARO obtains an IoI of 0.601 and IoR of 469.4 on Steam, and 1.976 and 1324.2 on Amazon Book. This demonstrates the superiority of our Bias and Reward Optimization framework in enhancing long-term user interest and item exposure. Moreover, BARO achieves the highest Acceptance score across both datasets (0.664 on Steam and 0.611 on Amazon Book), showing the importance of using reward estimation and bias mitigation to align recommendations with user preferences and expectations. For the metric of Coherence, it also obtains the best Coherence score on Steam (0.601), and a competitive score on Amazon Book (0.617), comparable to BiLLP (0.604) and T-PRA (0.629), indicating that the two-stage optimization framework enables BARO to maintain semantic consistency across multiple recommendation turns, while aligning the generated content with both user intent and long-term engagement goals.

Table 3: Ablation study results on Steam and Amazon Book datasets.

| Method | Steam | | | | | Amazon Book | | | | |
|---|---|---|---|---|---|---|---|---|---|---|
| | IoI | IoR | Acc. | Acce. | Coh. | IoI | IoR | Acc. | Acce. | Coh. |
| T-PRA | 0.584 | 432.8 | 0.894 | 0.588 | 0.403 | 1.783 | 1276.5 | 0.773 | 0.589 | 0.629 |
| -BD | 0.492 | 379.1 | 0.727 | 0.499 | 0.383 | 0.840 | 1182.0 | 0.453 | 0.423 | 0.345 |
| -RE | 0.599 | 399.2 | 0.778 | 0.640 | 0.560 | 1.481 | 1321.3 | 0.776 | 0.594 | 0.651 |
| -BD, RE | 0.360 | 311.9 | 0.664 | 0.308 | 0.305 | 0.686 | 738.5 | 0.381 | 0.354 | 0.217 |
| -SFT | 0.540 | 415.3 | 0.716 | 0.451 | 0.583 | 1.463 | 892.9 | 0.295 | 0.653 | 0.639 |
| BARO (ours) | 0.601 | 469.4 | 0.886 | 0.664 | 0.601 | 1.976 | 1324.2 | 0.802 | 0.611 | 0.617 |

## 5.2 ABLATION STUDY

This section presents the ablation study. `-BD` refers to the version of BARO without the bias detector. Similarly, `-RE` refers to BARO without the reward estimator. `-BD, RE` represents that both the bias detector and the reward estimator are removed, and the meta-controller directly decides whether to call ReGen or the action grounder. `-SFT` denotes the variant that the action grounder is not trained with SFT. The corresponding results are shown in Table 3. First, removing the bias detector (`-BD`) consistently hurts performance across almost all metrics, particularly Accuracy (Acc.) on both datasets, confirming that explicitly detecting and correcting popularity bias is crucial for improving recommendation quality. Second, dropping the reward estimator also leads to significant drops, especially in Coherence and Acceptance, indicating that

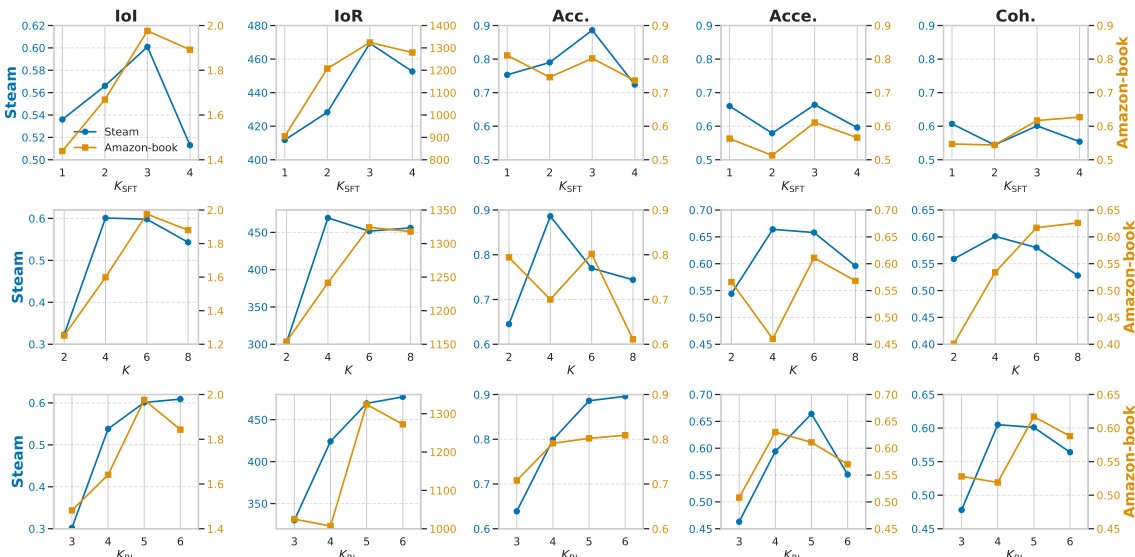

Figure 2: Hyperparameter sensitivity analysis for the number of SFT epochs ($K_{\text{SFT}}$), maximum tool calls ($K$), and RL epochs ($K_{\text{RL}}$) on the Steam and Amazon-book datasets.

refining noisy world-model rewards improves not only accuracy but also the trajectory consistency perceived by users. Notably, the joint removal of both tools (-BD, RE) yields the most severe degradation, highlighting their complementary roles in addressing distinct yet coupled challenges of bias and reward in offline recommendation. Third, eliminating the SFT initialization (-SFT) reduces overall performance, with a particularly sharp decline in Accuracy on Amazon Book, suggesting that the reward-distilled warm-up is essential for stabilizing subsequent RL training. Finally, the full BARO model achieves the best overall results, outperforming the T-PRA baseline in both IoI and IoR, while simultaneously improving Acceptance and Coherence. These results **validate the effective contributions of each component in BARO**.

### 5.3 ROBUSTNESS OF HYPERPARAMETERS

In BARO, we investigate the impact of three key factors: the number of SFT epochs $K_{\text{SFT}}$, the number of RL epochs $K_{\text{RL}}$, and the maximum number of tool calls $K$. The results in Figure 2 demonstrate clear trends. For SFT epochs (top row), we observe that a moderate number (around 3 epochs) yields the best balance across metrics, as too few epochs underfit while excessive tuning causes overfitting. For RL epochs (bottom row), performance steadily improves up to about 5 epochs and then saturates, confirming that policy refinement benefits from additional interactions but plateaus once the agent stabilizes. Regarding the tool budget $K$ (middle row), we find that allowing 4–6 calls provides consistent gains by enabling sufficient refinement, while an overly large $K$ leads to unnecessary regenerations and degraded efficiency. These trends highlight the importance of **training and inference budgets to maximize BARO's benefits without introducing instability or redundancy**.

## 6 CONCLUSION

In this work, we presented BARO, a meta-controlled and tool-augmented LLM agent tailored for recommender systems. Unlike prior ReAct-style agents that primarily enhance memory and planning, BARO explicitly addresses two fundamental challenges in offline recommendation: **bias mitigation** and **reward calibration**. By integrating a bias detector, a reward estimator, and an action grounder under the governance of a meta-controller, BARO injects debiasing and reward refinement directly into the decision-making loop.

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

# A APPENDIX

## A.1 STATEMENT ON LLM USAGE

Other than the LLM backbones leveraged in the experiments, the only LLM we used was ChatGPT-5, solely for manuscript polishing and grammar correction.

## A.2 DEMONSTRATION OF PROMPTS

| **Prompt template for Coarse Recommender** | |
| --- | --- |
| **Instruction:** | You are a recommendation expert. You should rely on the user's interaction history, the current regeneration round, and guidance from other modules. The goal is to recommend one item that serves as the most representative item you want to recommend at the current step. |
| **Input:** | The user's interaction history is $\{h_t\}$(END OF HISTORY).
It is your {#ReGen} times to generate an item in the current step.
The guidance information from upstream modules is: {g}. |
| **Response:** | Recommend the [item]. |

Figure 3: Prompts to generate an item with Coarse Recommender.

| **Prompt template for Meta-controller** | |
| --- | --- |
| **Instruction:** | You are the meta-controller of the recommendation pipeline. Your task is to evaluate the candidate set based on three key factors: popularity ratio, reward scale, and reward uncertainty.
- If the set appears strongly biased, call the Bias Detector for further examination.
- If rewards are very low or uncertainty is high, call the Reward Estimator for further examination.
- If the set quality is acceptable, call the Action Grounder to make the final recommendation.
In making decisions, also consider the user's interaction history and the current regeneration round.
Finally, provide a concise summary of the candidate quality to guide downstream modules. |
| **Input:** | Current candidate set is: {candidate set}(END OF CANDIDATE SET).
The corresponding statistical information of the candidate set is: $\{\mathcal{M}\}$(END OF STATISTICAL INFORMATION).
The user's interaction history is $\{h_t\}$(END OF HISTORY).
It is the coarse recommender's {#ReGen} times to generate a candidate set in the current step. |
| **Response:** | Call Bias Detector/Reward Estimator/Action Grounder, [summary text: $\mathcal{S}_{\text{meta}}$]. |

Figure 4: Prompts for tool-use and generating summary texts with Meta-controller.

| **Prompt template for Bias Detector** | |
|---|---|
| **Instruction:** | You are the Bias Detector. Your role is to carefully evaluate whether the candidate set contains excessive popularity bias. Use the provided candidate set, the user's interaction history, the regeneration count, and the diagnostic summary from the meta-controller to make your decision.
- If the bias is severe, call ReGen and provide a short summary to guide the coarse recommender.
- If the bias is acceptable, call the Action Grounder and provide a short summary to guide it. |
| **Input:** | The candidate set is {candidate set}(END OF CANDIDATE SET).
The corresponding statistical information of the candidate set is: $\{\mathcal{M}\}$(END OF STATISTICAL INFORMATION).
The user's interaction history is $\{h_t\}$(END OF HISTORY).
It is the coarse recommender's {#ReGen} times to generate a candidate set in the current step.
The diagnostic summary from the meta-controller is $\{\mathcal{S}_{\text{meta}}\}$. |
| **Response:** | - Either: call ReGen [diagnostic summary: $\mathcal{S}_{\text{bd}}$].
- Or: call Action Grounder [diagnostic summary: $\mathcal{S}_{\text{bd}}$]. |

Figure 5: Prompts to evaluate the candidate set's bias with Bias Detector.

| **Prompt template for Reward Estimator** | |
|---|---|
| **Instruction:** | You are the Reward Estimator. Your role is to check whether the candidate set suffers from low rewards or high uncertainty. Use the provided candidate set, the user's interaction history, the regeneration count, and the diagnostic summary from the meta-controller to make your decision.
- If the reward signals are unreliable, call ReGen and provide a short summary to guide the coarse recommender.
- If the signals are acceptable, call the Action Grounder and provide a short summary to guide it. |
| **Input:** | The candidate set is {candidate set}(END OF CANDIDATE SET).
The corresponding statistical information of the candidate set is: $\{\mathcal{M}\}$(END OF STATISTICAL INFORMATION).
The user's interaction history is $\{h_t\}$(END OF HISTORY).
It is the coarse recommender's {#ReGen} times to generate a candidate set in the current step.
The diagnostic summary from the meta-controller is $\{\mathcal{S}_{\text{meta}}\}$. |
| **Response:** | - Either: call ReGen [diagnostic summary: $\mathcal{S}_{\text{re}}$].
- Or: call Action Grounder [diagnostic summary: $\mathcal{S}_{\text{re}}$]. |

Figure 6: Prompts to assess the candidate set's rewards with Reward Estimator.

| **Prompt template for Action Grounder** | |
|---|---|
| **Instruction:** | You are the Action Grounder. Your task is to make the final recommendation from the candidate set. Use the user's interaction history, the regeneration count, and the diagnostic summaries from upstream modules to select the single best item. The goal is to maximize user satisfaction while avoiding hallucination. |
| **Input:** | The candidate set is {candidate set}(END OF CANDIDATE SET).
The user's interaction history is $\{h_t\}$(END OF HISTORY).
It is the coarse recommender's {#ReGen} times to generate a candidate set in the current step.
The diagnostic summary is from the Meta-Controller/Bias Detector/Reward Estimator, and the corresponding (diagnostic) summary is $\{\mathcal{S}_{\text{meta}}/\mathcal{S}_{\text{bd}}/\mathcal{S}_{\text{re}}\}$. |
| **Response:** | Recommend the [item]. |

Figure 7: Prompts for final recommendation with Action Grounder.

## A.3 PSEUDO CODE

The pseudo-code of BARO is provided here for reproducibility.

---

**Algorithm 1:** BARO's Training and Inference

---

**Input:** Offline logs $\mathcal{D}$, world model $\hat{R}$, coarse recommender $f_{CR}$, bias detector (BD), reward estimator (RE), action grounder $\pi_\theta$, meta-controller $\pi_\psi^{\text{meta}}$

**Output:** Trained BARO agent

**Phase I: Supervised Fine-Tuning (SFT)**

**for** *each batch $(s_t, \mathcal{I}_t)$ sampled from $\mathcal{D}$* **do**

    Query $\hat{R}$ to obtain $\{\widetilde{R}(s_t, i)\}, \{u(s_t, i)\}$

    Construct soft label distribution $y_i$ via Eq. 10

    Compute loss $\mathcal{L}_{\text{KD}}(\theta)$ with uncertainty weighting (Eq. 11)

    Update $\pi_\theta$ with LoRA adapters; keep $\pi_\psi^{\text{meta}}$, BD, and RE frozen

**Phase II: Reinforcement Learning (RL)**

**for** *episode = 1 to N* **do**

    Initialize history $h_t$, candidate set $\mathcal{I}_t \leftarrow f_{CR}(h_t, \#\text{ReGen}, g)$

    **for** *step = 1 to K* **do**

        Meta-controller selects $a_{\text{meta}} \in \{\text{BD, RE, AG}\}$

        **if** $a_{meta} = BD$ **then**

            BD evaluates $\mathcal{I}_t$ and outputs $\mathcal{S}_{\text{bd}}$

            If severe bias $\rightarrow$ regenerate $\mathcal{I}_t$ via $f_{CR}$

        **else if** $a_{meta} = RE$ **then**

            RE evaluates reward status and outputs $\mathcal{S}_{\text{re}}$

            If reward unacceptable $\rightarrow$ regenerate $\mathcal{I}_t$ via $f_{CR}$

        **else**

            Invoke action grounder $\pi_\theta$ to select final item $a_t$

            Break

        Compute meta reward $r_t^{\text{meta}}$ via Eq. 14

        Update $\pi_\psi^{\text{meta}}$ with PPO

    Compute action grounder reward $r_t^{\text{ag}}$ via Eq. 12

    Update $\pi_\theta$ with PPO

    Alternate updates between $\pi_\theta$ and $\pi_\psi^{\text{meta}}$

**Inference**

Given $h_t$, coarse recommender produces $\mathcal{I}_t$

Meta-controller inspects $\mathcal{M}_t$ and chooses BD, RE, or AG

If BD/RE $\rightarrow$ refine $\mathcal{I}_t$ and update summaries

Action grounder selects final recommendation $a_t$ from $\mathcal{I}_t$

Append $\mathcal{S}_{\text{meta}}, \mathcal{S}_{\text{bd}}, \mathcal{S}_{\text{re}}$ to ensure interpretability

---

