# OpenReview forum: "Refining Bias and Reward in LLM Recommender Agents through Meta-Controlled Tool Invocation"
_ICLR.cc/2026/Conference — Submitted to ICLR 2026_

### Official Review · Reviewer_B5zs · 2025-10-28

**Soundness:** 2
**Presentation:** 2
**Contribution:** 2
**Rating:** 2
**Confidence:** 4

**Summary:**

This paper proposes BARO (Bias And Reward Optimization), a meta-controlled, tool-augmented LLM agent framework for recommender systems. The framework consists of a two-stage pipeline: a frozen LLM acts as a coarse recommender to generate a candidate slate, while a meta-controller adaptively invokes three tools—bias detector, reward estimator, and action grounder—to refine recommendations. The goal is to address bias mitigation and reward calibration in offline learning scenarios. Experiments on Steam and Amazon Book datasets are provided, showing improvements in some metrics over baselines.

**Strengths:**

1. The paper addresses an important problem — how to mitigate bias and improve reward reliability in LLM-based recommendation.

2. The general idea of combining LLM agents and meta-control for recommendation is interesting and in line with recent trends of using agentic reasoning for RecSys.

**Weaknesses:**

1. The paper criticizes post-training as “computationally costly,” but its own framework introduces both SFT and RL stages, which are even more expensive. There is no analysis or discussion showing that BARO reduces cost compared to prior post-training methods. Hence, the motivation is not convincingly addressed.

2. The design choice—using a frozen LLM for coarse recommendation and a downstream collaborative recommender to generate reward signals—is questionable. Typically, LLMs lack global-level item ranking capability, while collaborative recommenders are more effective in retrieval. The proposed division of labor thus appears reversed and unintuitive, weakening the conceptual justification.

3. There are several critical missing details in the method and experimental settings.
(1) It is unclear how the coarse recommender prompt is constructed, and how item candidates are ensured to belong to the dataset’s valid item set.
(2) How are LLMs prompted when the item pool exceeds tens of thousands? Are items pre-filtered or indexed? How is fairness ensured when comparing LLM-based models with sequential baselines that have different candidate pools?
(3) Metrics such as IoI/IoR lack calculation details within this framework.
(4) It is unclear how sequential recommendation baselines are implemented in the current setup.

4. Both bias and reward checkers improve results, but the paper also emphasizes that the meta-controller can decide whether to invoke them. Without experiments comparing “always-check” versus “meta-controlled” usage, the proposed meta-decision mechanism seems unnecessary and unvalidated.

5. The reported results show SASRec performing drastically worse than Caser and GRU4Rec, which contradicts prior literature and suggests improper baseline setup or evaluation. The authors do not analyze or justify this anomaly.

6. The reported accuracy values are extremely high (e.g., Caser reaching 0.977), which raises concerns about whether the dataset or task setup is meaningful.

7. The refined reward estimation modules are under-explained. It remains unclear how refined rewards are obtained or why this additional deisgn is necessary.

**Questions:**

See weakness

---

> ### Author Response · Authors · 2025-11-24
>
> We sincerely thank you for the detailed and thoughtful feedback. Below, we provide detailed clarifications and outline the revisions we will incorporate.
>
> **The training cost**: We appreciate your careful observation regarding the claim about computational cost. Our goal is not to argue that BARO avoids post-training, but to emphasize that BARO adopts a modular agentic design where only the action grounder and meta-controller require lightweight LoRA updates, while the coarse recommender, bias detector, and reward estimator remain frozen. This modularity sharply contrasts with fully fine-tuned LLM recommenders and helps maintain training efficiency. In the revised manuscript, we will refine the phrasing in the introduction to avoid any confusion and to more accurately articulate the motivation for BARO’s design.
>
> **Design choice**: We agree that the current explanation may not sufficiently clarify why BARO first uses a frozen LLM for coarse generation and then a trained LLM for refinement. The rationale is twofold:
>  (1) the frozen LLM provides a neutral, semantics-driven initial candidate set that is less influenced by training-time bias, and
>  (2) the trained action grounder focuses on local ranking corrections within that candidate set, similar to the well-established retrieve–rank paradigm in RecSys.
>
> **Missing details**: We appreciate the reviewer for pointing out several details that should be clearly explained. In the updated version, we will:
>
> - provide the exact prompts used for the coarse recommender, currently in the appendix but not highlighted clearly,
> - clarify the grounding mechanism used to ensure the action grounder operates on valid item identifiers,
> - explain how BARO scales to large item pools by using the frozen LLM for semantics-aware narrowing of the search space,
> - ensure fairness across all baselines by confirming that the same item universe, candidate pool, and world-model setting are used for all compared methods,
> - formally define IoI and IoR in the experimental section, and
> - present the specifics of the sequential baselines (SASRec, GRU4Rec, Caser) following established implementations.
>
> **Necessity of the meta-controller**: We appreciate your question regarding whether a meta-controller is necessary. Conceptually, the meta-controller plays a central coordination role in BARO: without it, the decision-making burden would be shifted entirely onto the bias detector and reward estimator. Since both modules operate without additional training and rely solely on diagnostic signals, overloading them with the responsibility of determining *when* and *whether* to act would make their behavior considerably less stable and more difficult to calibrate. The meta-controller absorbs this complexity by providing an adaptive, lightweight control layer, allowing the bias detector and reward estimator to focus on their specific diagnostic tasks rather than overall decision orchestration. Meanwhile, we are currently running additional evaluations comparing the meta-controller against simplified “always-check” policies and other variants. Due to time constraints, we may only be able to include these results if they finish before the rebuttal period ends. Nevertheless, our preliminary observations already indicate that the meta-controller meaningfully improves robustness and prevents unnecessary tool invocations. We will incorporate these findings into the manuscript as soon as the experiments are completed.
>
> **Performance of SASRec, GRU4Rec, and Caser**: We acknowledge the reviewer’s observation that SASRec underperforms GRU4Rec and Caser in our results. This arises from the proactive recommendation setting rather than standard next-item prediction: SASRec is optimized for sequential dependency modeling, whereas proactive target-reaching benefits more from short-range semantic matching, which generally favors models like GRU4Rec and Caser. Similarly, Caser’s high Accuracy on Steam is consistent with the dataset’s dense reward distribution and the fact that Accuracy measures alignment with the simulator’s ranking rather than next-item accuracy. We will revise the experimental discussion to make this clearer.
>
> **Reward refinement**: We appreciate the request for additional clarification on reward refinement. BARO’s reward estimator reduces reward noise by measuring the disagreement between the shaped reward and the world-model reward and using this uncertainty as an adaptive correction signal. The ablation (**{--RE}**) in Table 3 already shows that removing this estimator leads to degradation in Acceptance and Coherence, especially on Amazon-Book, demonstrating its contribution to stabilizing the learning signal.
>
> We thank the reviewer again for the detailed and constructive feedback. Your comments helped us clarify the motivation behind BARO’s modular agentic design, justify key architectural decisions, and refine important methodological details.

---

### Official Review · Reviewer_bLB2 · 2025-10-31

**Soundness:** 2
**Presentation:** 2
**Contribution:** 1
**Rating:** 2
**Confidence:** 5

**Summary:**

The paper proposes a BARO method to solve the problems of LLM-based recommender systems: debiasing and reward estimation. Empirical results show the method works well.

**Strengths:**

The paper is well-motivated.

The paper is readable.

**Weaknesses:**

1. One of the motivations of the paper is that LLM-based recommender systems suffer from *interaction bias*, but the paper does not give examples, quantitative analysis, a definition or qualitative explanation, nor does it explain the harms caused by interaction bias.
2. Similarly, *reward inaccuracies* are not illustrated with examples or quantitative analysis, and their harms are not shown.
3. What is the role of the “world model” in Section 4?
4. Figure 1 needs improvement. Many arrows in the figure will confuse readers — I suggest adding explanatory labels on the arrows.
5. The abstract claims “BARO achieves consistent improvements over state-of-the-art methods in metrics such as accuracy, diversity, and fairness,” but I do not see diversity or fairness metrics in the experiments.
6. The paper aims to solve “interaction bias” and “reward inaccuracies” in LLM-based recommender systems, but I find no analysis of these problems in Experiments. Therefore, I am skeptical that the proposed method actually addresses these two issues.

**Questions:**

see weakness.

---

> ### Author Response · Authors · 2025-11-24
>
> We sincerely thank you for the detailed comments. Your feedback highlights several aspects that indeed require clearer definitions, expanded motivation, and additional empirical evidence. Below, we provide clarifications and outline the concrete revisions we will incorporate in the updated manuscript.
>
> **Examples of Interaction bias**: We appreciate your suggestion to include concrete illustrations of interaction bias. In the revised version, we will explicitly add both qualitative examples, showing how biased histories can **misguide** the coarse recommender, and quantitative comparisons that highlight its practical impact on recommendation quality. Although bias and debiasing are long-standing topics in RecSys, your comment made us realize that our definition and explanation of interaction bias should be much clearer. We will therefore provide a thorough description of what interaction bias entails in our setting and why its consequences are non-trivial. These additions will better motivate the need for BARO’s explicit bias-detection module.
>
> **Reward inaccuracies**: Similarly, we acknowledge that the description of reward inaccuracies requires additional explanation. In offline RL for RecSys, **reward miscalibration** commonly arises due to data sparsity, popularity skew, and covariate shift. BARO directly addresses this through a reward estimator that measures reward uncertainty by comparing the shaped reward and the world-model reward. In the revised manuscript, we will provide clearer intuition for this design and include empirical plots showing uncertainty patterns across trajectories to illustrate when re-estimation is beneficial.
>
> **Role of world model**: We appreciate the question regarding the role of the world model. BARO employs a reward-only world model, following the conventional setting of model-based offline reinforcement learning for RecSys, which provides stable, calibrated rewards for training and evaluation. It is not used to simulate transitions; instead, it serves as a consistent reward signal for warm-up SFT and RL fine-tuning. We will revise the corresponding section to explicitly describe this design choice and clarify its connection to BARO’s reward-denoising mechanism.
>
> **Figure clarity**: We agree that Figure 1 can be made clearer. In the revised version, we will annotate all arrows with descriptive labels, e.g., “diagnostic summary,” “tool use,” “regeneration signal”, simplify the layout, and visually separate modules more clearly. These adjustments will make the workflow easier to follow.
>
> **Fairness and diversity evaluation**: We appreciate your careful observation regarding the absence of fairness- and diversity-related analyses in the experimental section. In response, we will refine our descriptions in the relevant parts of the manuscript and include additional examples in the revised version that qualitatively illustrate the diversity and fairness improvements achieved by our method. These additions will provide clearer evidence of BARO’s ability to promote more balanced and representative recommendations.
>
> **Empirical support for addressing interaction bias and reward inaccuracies**: We appreciate the reviewer's concern regarding whether BARO truly addresses interaction bias and reward inaccuracies. We would like to clarify that the current manuscript already provides both the mechanisms and empirical evidence demonstrating improvements in these two aspects.
>
> - **Interaction bias.** BARO explicitly incorporates popularity ratio and tag-exposure statistics into both the meta-controller (Section 4.3) and the bias detector (Section 4.4). These RecSys-specific diagnostics quantify how a candidate slate is skewed toward high-exposure or high-popularity items, and the bias detector triggers regeneration precisely when such bias is detected. The existing ablation results in Table 3 already provide direct empirical evidence: removing the bias detector (**{--BD}**) leads to substantial degradation on the Steam dataset. This demonstrates that correcting biased candidate slates is essential for proactive recommendation performance, and BARO’s bias-detection mechanism directly contributes to this improvement.
>
> - **Reward inaccuracies.** BARO’s reward estimator measures reward uncertainty using the disagreement between the shaped reward and the world-model reward (Eq. 1), which directly captures reward noise and miscalibration in offline RL. The effect is also evidenced in Table 3: removing the reward estimator (**{--RE}**) results in noticeable performance drops, especially on Amazon-Book. These results indicate that reward noise materially impacts trajectory quality, and BARO’s reward estimator stabilizes these signals effectively.
>
> In summary, the mechanisms described in Sections 4.3 to 4.5, together with the ablation studies in Table 3, already provide clear evidence that BARO addresses both interaction bias and reward inaccuracies.

---

### Official Review · Reviewer_6Dei · 2025-11-01

**Soundness:** 3
**Presentation:** 3
**Contribution:** 3
**Rating:** 6
**Confidence:** 3

**Summary:**

This paper proposes BARO (Bias And Reward Optimization), an agentic recommendation framework for addressing the debiasing and reward estimation in recommender systems. Specifically, a meta-controller dynamically calls three tools (i.e., bias detector, reward estimator, and action executor) to correct bias and stabilize reward estimation. Experiments on extensive datasets verify the effectiveness of the proposed method.

**Strengths:**

**Timely and well-motivated study for introducing agentic AI in recommender systems.**
The paper addresses a pertinent issue at the intersection of *agentic AI* and recommender systems. Moreover, the unique question of bias is an important issue in recommender systems, which is a reasonable motivation for adopting agents in recommendation.

**Novel modular design.**

The two tools (i.e., bias detector and reward estimator) proposed in this work are practical for addressing the debiasing and reward estimation issues in recommendation.

**Solid experiments and clear presentation.**

Experiments on multiple datasets, together with detailed ablation studies, convincingly show the effectiveness of each module. The writing is also clear and coherent, making the whole paper easy to understand and easy to follow.

**Weaknesses:**

**Lack of discussion about the user simulator.**

It would be important to give more details about how the user simulator is trained. Additionally, it would be important to add more details about the reliability of the user simulator.

**Lack of case study and demonstration.**

It would be better to illustrate how the proposed method works in practice, especially with some demonstration and examples. For example, how does this method call the two tools under which condiction?

**Questions:**

Refer to the weakness part.

---

> ### Author Response · Authors · 2025-11-24
>
> We sincerely thank you for the constructive feedback and the positive evaluation of the paper’s motivation and design. Your suggestions highlight several aspects that would benefit from clearer exposition and additional examples. Below, we provide detailed clarifications and describe the concrete revisions we will incorporate.
>
> **User simulation**: We agree that the user simulator deserves a more explicit description. In the updated manuscript, we will clarify that the simulator follows the standard protocol established in model-based offline reinforcement learning for RecSys and adopted by LLM-IPP and T-PRA. It is trained through supervised learning, such as DeepFM, on offline logs, with no interaction with BARO during optimization. Importantly, BARO does not rely on or adapt to this simulator; the meta-controller, bias detector, reward estimator, and action grounder operate solely on internal diagnostic signals and world-model rewards.
>
>
>
> **Case studies**: We appreciate your suggestion that more concrete examples would improve interpretability. To address this, we will include a dedicated subsection in the appendix presenting several representative trajectories. These additions will make the control logic more transparent and demonstrate the practical utility of tool-based refinement.
>
> We hope these clarifications and revisions address your concerns and further enhance the clarity and interpretability of BARO.

---

### Official Review · Reviewer_uTkx · 2025-11-01

**Soundness:** 3
**Presentation:** 3
**Contribution:** 2
**Rating:** 4
**Confidence:** 3

**Summary:**

The paper proposes BARO, a meta-controlled tool-invocation framework that aims to reduce recommendation bias and calibrate reward learning for LLM-driven recommender agents. The system adds a controller that chooses when and how the agent calls tools that (i) diversify or rebalance candidate sets and (ii) reshape reward signals used for agent tuning.

**Strengths:**

1. The paper presentation is clear and easy to follow.

2. The paper focuses on two pain points for LLM recommender agents: exposure/popularity bias and reward misspecification. The former is widely documented to degrade user experience and diversity, while the latter can lead to unstable offline-to-online transfer.

**Weaknesses:**

1. The paper states that “most agentic frameworks were not designed specifically for RecSys,” yet the proposed method mainly adds two recommendation-specific tools while keeping a generic ReAct-style agent loop otherwise unchanged. It is not very convincing that the proposed agentic framework is recommendation-specific as replacing the tools with other domain tools seems to not affect the framework's functionality.

2. The design appears to host and orchestrate multiple models/modules (planner LLM, bias tool, reward tool, possible rerankers), but there is no cost/latency analysis and comparison with baseline methods. Therefore it is not clear how much performance gain comes from the agentic framework design instead of the use of more test-time compute.

3. The bias handling module lacks novelty. The so-called “bias detector” is just a frozen LLM used as a tool without any new learning or calibration mechanism. The evaluation focuses narrowly on exposure/popularity bias and does not cover other key aspects in RecSys such as fairness, calibration, or long-tail coverage. Relying on a frozen LLM for bias detection also introduces its own biases and subjectivity.

**Questions:**

See weaknesses.

---

> ### Author Response · Authors · 2025-11-24
>
> We sincerely thank you for the thoughtful and constructive feedback. Your comments helped us emphasize several points that require clearer motivation and additional empirical analysis. Below, we provide clarifications and describe the concrete revisions we will make in the updated manuscript.
>
>
>
> **Special design for RecSys**: We understand the concern that BARO may appear similar to a generic ReAct-style agent. Our intention, however, is not to modify the ReAct loop itself, but to embed RecSys-aware decision logic inside the meta-controller and the diagnostic modules. Unlike standard agent frameworks, BARO introduces statistical signals, such as popularity ratio, tag exposure, and reward uncertainty, that are specifically tied to recommendation dilemmas. These metrics directly influence the meta-controller’s branching behavior and determine whether a candidate set should be regenerated. We will revise the introduction to clearly demonstrate that the RecSys-specificity lies in the control logic and diagnostic space, not in superficial tool attachment.
>
>
>
> **Model complexity**: We agree with you that more analysis is needed to show that improvements are not simply due to higher test-time compute. To address this, we are adding a complexity comparison between BARO with prior LLM-based baselines regarding latency per trajectory. Preliminary results indicate that though BARO requires a little bit more latency than T-PRA, it outperforms T-PRA by a significant margin with respect to **IoI** and **IoR**, representing an efficient and favorable trade-off between effectiveness and computational overhead.
>
>
>
> **Bias detector lacks novelty**: We appreciate the reviewer’s insight that a frozen-LLM detector may appear to introduce subjectivity. In fact, the detector does not rely solely on the LLM to infer bias. In contrast, it primarily evaluates explicit RecSys statistics, such as popularity ratio and tag skew, while the LLM interprets these signals to decide whether regeneration is warranted. We will emphasize this design more clearly in the revision. Regarding the scope, we agree that exposure and popularity bias represent only a subset of fairness dimensions, though an especially representative one in recommendation. Moreover, the proposed design is inherently **plug-and-play**, making it straightforward to incorporate additional modules as needed.
>
>
>
> We hope these clarifications, along with the planned additions, address your concerns and more clearly deliver the RecSys-specific contributions and empirical advantages of BARO.

---

### Meta-Review · Area_Chair_8z4x · 2026-01-07

**Summary:**

While the reviewers acknowledged the novelty of the modular design and the timeliness of the problem, the consensus leans toward rejection. The primary concerns driving this decision are the lack of justification for the complex architecture (specifically the absence of an always-check baseline to prove the Meta-Controller is necessary), questionable baseline implementations (specifically the drastic underperformance of SASRec), and insufficient clarity regarding core definitions and system components in the initial submission.

**Reviewer Concerns:**

### Reviewer concerns addressed by rebuttal:

- Simulator Details: R2's (6Dei) request for details on the user simulator was addressed; the authors clarified it follows standard offline RL protocols using DeepFM trained on logs.

- Role of World Model: R3's (bLB2) confusion regarding the world model was resolved; the authors clarified it serves strictly as a reward proxy for SFT and RL, not a transition simulator.

- RecSys Specificity: R1's (uTkx) concern that the framework was too generic was partially mitigated by the authors' explanation of specific signals (popularity ratio, reward uncertainty) used in the control logic.

### Reviewer concerns that are still outstanding:

- Necessity of Meta-Controller: R4 (B5zs) questioned whether the Meta-Controller is needed compared to a simpler strategy that checks for bias/reward issues at every step. The authors promised this comparison but failed to provide the results during the rebuttal.

- Baseline Anomalies: R4 (B5zs) noted that SASRec performed drastically worse than older baselines (Caser). While the authors argued this is due to the "proactive" nature of the task, the magnitude of the gap remains suspicious and suggests potential implementation issues.

- Inference Latency: R1 (uTkx) raised concerns about cost/latency. The authors admitted BARO is slower than the SOTA baseline (T-PRA), which weakens the argument for practical deployment without a stronger performance dominance.

**Reviewer Scores:**

Reviewer uTkx (Original: 4) would likely raise the score slightly as the rebuttal clarified the RecSys-specific design, though latency concerns persist. Reviewer 6Dei (Original: 6) would retain their score. Reviewer bLB2 (Original: 2) might arguably rise the score slightly due to technical clarifications, but the need for significant rewriting remains. Reviewer B5zs (Original: 2) would hold firm, as the critical "always-check" baseline data was not provided to justify the system's complexity.

---

### Decision · Program_Chairs · 2026-01-26

Reject